# ORACLEMAMBA: A DYNAMIC MARKET-GUIDED AND TIME STATE SELECTION FRAMEWORK FOR ROBUST STOCK PREDICTION

## ABSTRACT

Stock price prediction is a complex challenge due to the inherent volatility of financial markets and the influence of diverse factors such as macroeconomic conditions, capital flows, and market sentiment. Recent joint stock forecasting models focus on extracting temporal patterns from individual stock price series and combining them to model stock correlations. However, these models face two critical limitations: first, in long-term predictions, they retain both informative and excessive states, amplifying noise and increasing complexity; second, in short-term predictions, they prioritize market indices and technical indicators, neglecting the real-time influence of market sentiment, which can drive price movements independent of traditional indicators. While state space models (SSMs) like Mamba improve efficiency and capture long-distance relationships, they still underperform compared to Transformer-based models. To address these challenges, we propose OracleMamba, a novel framework that integrates a dynamic market-guided module for short-term forecasting and a SelectiveMamba module for long-term forecasting. The dynamic market-guided module fuses objective market data and subjective sentiment analysis to enhance short-term prediction accuracy. The SelectiveMamba module efficiently captures both spectral and temporal features using a 3D scan mechanism, which extracts and filters key signals from the time-series data. By integrating spectral features to identify market rhythms and temporal features to track price movements over time, the SelectiveMamba module reduces noise and preserves critical information for long-term forecasts. This framework significantly improves both model efficiency and accuracy, outperforming existing approaches across real-world stock prediction tasks.

## 1 INTRODUCTION

Stock price forecasting is crucial for informed investment decisions but remains challenging due to the volatile nature of financial markets. Unlike traditional time series data (Poudel et al., 2024; Yi et al., 2024; Dong et al., 2024; Wu et al., 2024; Chen et al., 2024; Li et al., 2024b), stock prices are influenced by diverse factors like macroeconomic conditions, capital flows, sentiment, and unforeseen events. This complexity creates interdependencies across stocks, making it hard to isolate individual movements without considering broader market dynamics. In the global economy, shifts in investor sentiment can quickly spread, further complicating accurate predictions.

Traditional models (Feng et al., 2019; Xu et al., 2021; Wang et al., 2021) in the stock prediction task rely on predefined correlation structures based on industry sectors, using static graphs to model relationships between stocks. Recent works (Li et al., 2024b; Yoo et al., 2021) in joint stock forecasting have primarily focused on extracting temporal patterns from individual stock price series and combining them to model stock correlations. However, these methods overlook high noises in stock prices. Moreover, they face two key limitations. First, for long-term predictions, recent works (Vaswani, 2017; Yoo et al., 2021; Li et al., 2024a) retain all informative and excessive states, amplifying noise and increasing complexity , which results in quadratic time complexity and suboptical performance because they struggle to distinguish informative features, incorporating the vast amount of noise in stock data, particularly in real-time applications. Second, for short-term predictions, previous works (Li et al., 2024a) solely focus on objective market indicators, such as market index

and technical indicators, neglecting the influence of subjective market sentiment, which reflects the real-time state of investors and markets and usually drive price movements that completely diverge from market index trends, especially in today's digital age, where public sentiment can rapidly shift market dynamics (due to social media, news, etc.). With the increasing interdependence of global markets, previous models are insufficient for capturing the short-term market fluctuations and responding to complex, volatile conditions. Although recent state space models(SSMs) like Mamba (Gu & Dao, 2023; Dao & Gu, 2024) are more efficient to train than RNNs and are more capable at handling long distance relationships, offering a structured approach to modeling time-series data, but directly incorporating SSMs still lag behind the performance of comparably sized Transformer-based models.

To tackle these challenges, this paper proposes a novel framework OracleMamba, featuring a dynamic market-guided module and a SelectiveMamba module. For short-term forecasting, the dynamic market-guided module integrates comprehensive feature representations to guide predictions, enhancing the model's accuracy. Specifically, the comprehensive feature representations integrates both objective market data and subjective sentiment analysis, adjusting the importance of each in real time to reflect evolving market conditions. For long-term forecasting, the SelectiveMamba module equips the model with contextual awareness at every temporal location, allowing it to efficiently identify and retain informative states. This module comprises three key components: a Time-Spectral State Space (TSSS) layer, a 3D scan layer, and a fusion layer. The Time-Spectral State Space (TSSS) module is a novel architecture designed to capture both temporal dependencies and spectral features in time-series data. By integrating Dynamic Temporal Extractors (DTE) for time-domain patterns and Dynamic Spectral Extractors (DSE) for frequency-domain characteristics, the TSSS effectively models the complexities of market behavior. DTE tracks evolving temporal patterns, while DSE identifies key spectral components such as cyclical trends. This dual-domain approach allows TSSS to simultaneously capture short-term fluctuations and long-term trends, providing a robust framework for forecasting non-stationary, multi-scale market dynamics. The 3D Scan Layer introduces an advanced approach for analyzing market data, capturing intricate interactions across three key dimensions: time, stock, and market state. Unlike conventional 1D or 2D approaches (Sherstinsky, 2020; Li et al., 2024a), which often overlook deeper interdependencies, this method systematically extracts features along all three axes. The time dimension traces historical trends, the stock dimension captures inter-stock correlations, and the market state dimension reflects broader economic influences. This hierarchical, multi-dimensional scanning approach significantly enhances the model's capacity to interpret complex market dynamics, providing a more precise and comprehensive foundation for stock prediction.

Our approach combines a dynamic market-guided module for enhanced short-term predictions by integrating real-time market sentiment and data, with the SelectiveMamba module, which filters noise and preserves key signals for strong long-term correlations. This balance improves accuracy by addressing both short-term volatility and long-term stability.

To conclude, our research presents several key contributions:

- We propose a comprehensive framework that seamlessly integrates the dynamic market-guided module and the SelectiveMamba module, effectively tackling the challenges posed by noisy data and enhancing both short- and long-term stock price predictions.

- We are the first to introduce Mamba into stock price forecasting, harnessing its potential to adeptly manage complex, noisy time-series data. By enhancing Mamba, the Selective-Mamba module is empowered to capture features across both spectral and temporal domains. Our innovative 3D scan method facilitates the comprehensive extraction of features and the interplay between them.

- In contrast to prior models that focus exclusively on objective factors, we incorporate subjective factors by integrating market sentiment, allowing us to effectively capture the impact of emotional dynamics on stock predictions.

- Our approach achieves state-of-the-art performance across multiple challenging datasets, demonstrating its robustness and effectiveness in stock price forecasting. Comprehensive experiments validate the design choices of our proposed method, consistently outperforming established baselines and highlighting its superiority in capturing complex market dynamics.

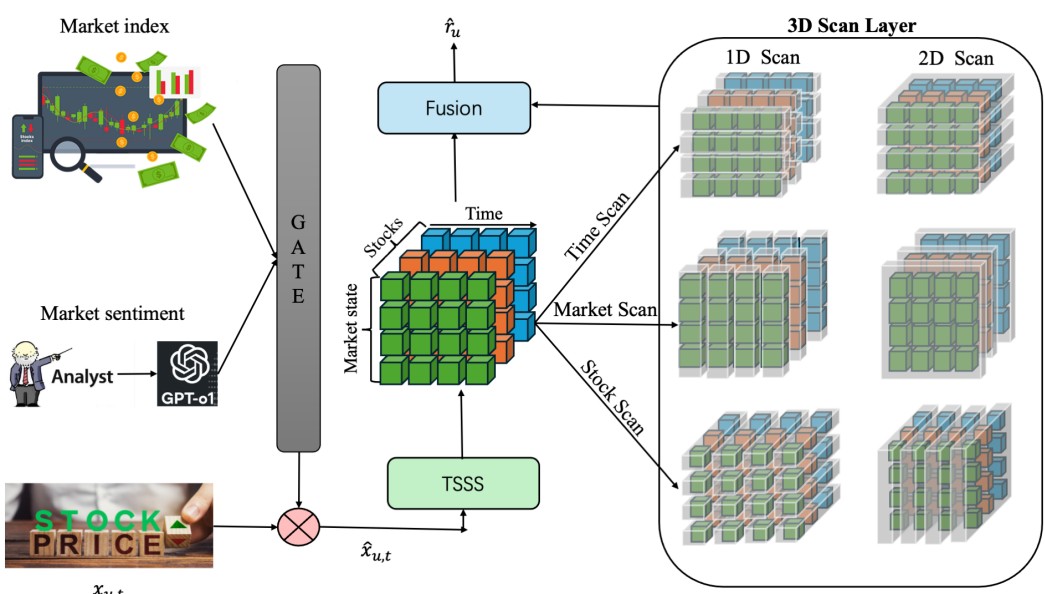

Figure 1: Overview of our framework

## 2 RELATED WORK

### 2.1 TIME-SERIES FORECASTING WITH DEEP LEARNING

Traditional deep learning models for time-series forecasting Li & Pan (2022); D'Amato et al. (2022); Li et al. (2024a), such as Long Short-Term Memory (LSTM) networks Yu et al. (2019) and Gated Recurrent Units (GRU) Cho (2014), have been extensively applied to stock price prediction. These models capture sequential dependencies by maintaining hidden states that evolve over time. However, they suffer from several limitations, including difficulty in learning long-range dependencies, susceptibility to vanishing gradient problems, and high computational costs due to their sequential nature. More recently, Transformer-based architectures Vaswani (2017) have been applied to time-series prediction tasks due to their superior ability to capture long-range dependencies through self-attention mechanisms. Models such as the Temporal Fusion Transformer (TFT) Lim et al. (2021) and the Informer Zhou et al. (2021) have demonstrated improved performance in multivariate time-series forecasting by leveraging self-attention to model complex dependencies across time and feature dimensions. However, Transformers scale quadratically with sequence length, making them computationally expensive and less suitable for applications requiring real-time or high-frequency updates, such as stock trading (Katharopoulos et al., 2020; Choromanski et al., 2020; Kitaev et al., 2020).

### 2.2 STATE-SPACE MODELS AND STRUCTURED MATRICES

State-space models (SSMs) have recently gained significant attention as a promising alternative for efficient sequence modeling, particularly in tasks requiring the handling of long-range dependencies. For example, the Structured State Space for Sequence Modeling (S4) Gu et al. (2021) has demonstrated that SSMs can achieve linear-time complexity with respect to sequence length by capitalizing on the properties of structured state-space representations. This makes them well-suited for tasks such as financial forecasting. However, despite the advances seen in models like Mamba Gu & Dao (2023) and Mamba-2 (Dao & Gu, 2024), SSMs still face limitations in terms of expressiveness and flexibility compared to Transformer-based models. While these SSMs may excel at selecting state spaces in the time domain, they all overlook the richer periodic features that can be exploited from the frequency domain, which further hampers their ability to capture complex patterns across different sequences. As a result, SSMs continue to struggle in fully matching the capabilities of Transformers when it comes to modeling highly expressive and diverse sequential data.

## 3 METHODS

**Problem Formulation.** At each time step $t \in [1, \tau]$, we collect indicators for each stock $u \in S$ to construct a feature vector $\mathbf{x}_{u,t} \in \mathbb{R}^F$, where $S$ is the set of stocks, and $F$ is the dimensionality of the feature space. Building on established research in stock market prediction (Li & Pan, 2022; D'Amato et al., 2022; Li et al., 2024a), our goal is to predict the relative change in stock prices rather than their absolute values. We define the return ratio $r_u$ for stock $u$ as follows:

$$r_u = \text{Norm} \left( \frac{c_{u,\tau+d} - c_{u,\tau+1}}{c_{u,\tau+1}} \right), \tag{1}$$

where $c_{u,t}$ is the closing price of stock $u$ at time $t$, and $d$ represents a predefined prediction interval. This return ratio normalizes price movements across different stocks, facilitating comparison of relative performance rather than focusing on absolute price changes. Since investment strategies typically aim to rank and select stocks with the highest expected returns, we adhere to TSF standards by applying normalization and de-normalization techniques to both input and output, effectively addressing distribution shift issues (Yang et al., 2020; Liu et al., 2022).

**Market State Encoding.** To create a comprehensive understanding of the market's current dynamics, we introduce a vector representation $m_t$, which integrates data from two distinct perspectives: objective market metrics and subjective insights. (1) The subjective market context is derived from analyst projections. Using the GPT-O1 model, we process textual data from analyst reports and other financial documents spanning diverse industries and regions. The model synthesizes this input to produce sentiment estimates that encapsulate expert views on upcoming market developments. (2) The objective market context is captured by three principal factors: the market index level, reflecting a weighted average of key stocks $S'$, proportional to their market capitalization, providing a broad indicator of market trends; the overall trading volume, which quantifies market liquidity and investor participation; and capital movement trends, which track shifts in funds across different sectors, offering insights into investor sentiment and projections for future market conditions. By combining these objective metrics with subjective expert assessments, our model constructs a well-rounded market context vector $m_t$ that represents both current market states and anticipates potential trends, improving the accuracy of stock predictions across sectors and regions.

**Definition (Market-Guided Stock Forecasting).** Given the historical stock data $\{x_{u,t}\}_{u \in S, t \in [1,\tau]}$ and the market context vector $m_\tau$, market-informed stock price forecasting is the task of predicting the normalized return rates $\{r_u\}_{u \in S}$ for the future.

**Overview.** Figure 1 illustrates the architecture of our proposed method, OracleMamba, which comprises four key components. (1) Market-Guided Gating. We construct a vector representing the market status by integrating objective market data and sentiment analysis. This vector is used in the gating mechanism to adjust feature importance dynamically, enabling real-time feature selection. (2) Time-Spectral Aggregation. At each time step, the Time-Spectral State Space (TSSS) layer aggregates temporal and spectral features, capturing both short-term fluctuations and long-term trends. (3) 3D Market Scan. The 3D scan layer captures interactions across three dimensions: time, stock, and market state. This multi-dimensional scan enhances feature extraction, preserving essential signals while reducing noise. (4) Fusion Module. Fusion module integrate the features from the TSSS and 3D scan layers, refining the final representation for prediction. The resulting comprehensive stock embedding is passed to the prediction layers for stock price forecasting. We discuss each step in more detail in the following subsections.

**Feature Modulation Mechanism.** We propose a feature modulation approach that dynamically adjusts the importance of each stock feature based on the market context. This method assigns weighting coefficients to features, enabling the model to emphasize or suppress attributes as needed. The model learns to scale these coefficients optimally during training, improving prediction accuracy by focusing on the most relevant features. To align the market context vector $m_\tau$ (dimension $F'$) with the stock feature space $F$, a linear transformation is applied. A Softmax function then generates weighting coefficients:

$$\alpha(m_\tau) = F \cdot \text{softmax}_\beta(W_\alpha m_\tau + b_\alpha), \tag{2}$$

where $W_\alpha$ and $b_\alpha$ are learnable parameters, and $\beta$ controls the focus of the distribution. A lower $\beta$ sharpens the focus on specific features, while a higher $\beta$ distributes emphasis more evenly. These modulation weights $\alpha(m_\tau)$ apply uniformly across all stock feature vectors $\{x_{u,t}\}_{u \in S, t \in [1,\tau]}$, ensuring consistent feature selection based on market conditions. The final adjusted feature vectors are:

$$\tilde{x}_{u,t} = \alpha(m_\tau) \circ x_{u,t}, \tag{3}$$

where $\circ$ represents element-wise multiplication, aligning features with the current market environment to enhance forecasting accuracy.

**Time-Spectral State Space (TSSS).** The Time-Spectral State Space (TSSS) is designed to capture both temporal and spectral features by integrating dynamic spectral extractors(DSE) and dynamic temporal extractors (DTE). This allows TSSS to effectively process time-domain dependencies and frequency-domain characteristics, providing a comprehensive understanding of complex time series data. TSSS is parameterized by four key components $(\Delta, A, B, C)$, which define a sequence-to-sequence transformation that operates in two stages: discretization of temporal dynamics and computation of spectral interactions. This structure enables the model to capture both short-term variations and long-term dependencies in the data. In the TSSS architecture, the DTE performs the initial feature transformation, focusing on temporal information. Meanwhile, the DSE derived from the dynamic spectral operator, captures frequency-domain characteristics, ensuring the model recognizes the underlying spectral patterns:

$$
\begin{aligned}
\text{DSE} &= C e^{A(t-s)} B \\
\text{DTE} &= \sigma(W_o \cdot [h_{t-1}, x_t] + b_o) \cdot \tanh\left(\sigma(W_f \cdot [h_{t-1}, x_t] + b_f) \cdot C_{t-1}\right. \\
&\quad + \sigma(W_i \cdot [h_{t-1}, x_t] + b_i) \cdot \tanh(W_C \cdot [h_{t-1}, x_t] + b_C)
\end{aligned}
\tag{4}
$$

where $h_t$ is the hidden state at time $T$. To further enhance its ability to focus on meaningful patterns, a fusion module is introduced. This mechanism dynamically fuses the transformed spectral information derived from the DSE with temporal features from the DTE, allowing the model to emphasize relevant spectral and temporal patterns while filtering out noise:

$$\tilde{x}_{u,t}^{\text{TSSS}} = \text{Fusion}(\text{DSE}^{\text{TSSS}}(\tilde{x}_{u,t}), \text{DTE}^{\text{TSSS}}(\tilde{x}_{u,t})) \tag{5}$$

By incorporating a fusion module, TSSS captures essential temporal and spectral features and enhances the interaction between these two domains. This enables more accurate and robust predictions, as the model can adaptively balance the processing of time-series and frequency-domain data.

**3D Scan Layer.** To comprehensively capture interactions across the time, stock, and market state dimensions, we introduce a 3D scanning mechanism. This mechanism systematically extracts features along these three axes, enabling the model to capture intricate dependencies that simpler 1D or 2D scanning approaches miss. First, a 1D scan is applied independently to each dimension—time, stock, and market state. This step allows the model to process each dimension as an ordered sequence. The 1D scan for the time dimension focuses on historical price movements, capturing temporal patterns and trends. Similarly, the 1D scan for the stock dimension processes stock-specific information, detecting inter-stock dependencies, while the 1D scan for the market state dimension isolates macroeconomic or market-wide conditions, such as overall capital flows or trading volumes. After the 1D scan, a 2D scan is performed to model interactions between pairs of dimensions. Specifically, $2\text{DScan}_{\text{Time}}$ explores relationships between consecutive time steps, identifying temporal dependencies and correlations. The $2\text{DScan}_{\text{Stock}}$ captures co-movement patterns between different stocks, enhancing the stock-specific feature representations. Finally, $2\text{DScan}_{\text{Market}}$ models how different market regimes affect stock behaviors over time, enriching the representation of macro-level market conditions. The scanning process is applied hierarchically across all three dimensions: a 1D scan operates within each dimension independently to capture sequential characteristics and a 2D scan then combines pairs of dimensions to extract cross-dimensional interactions. Formally, the 3D scan process is described as follows:

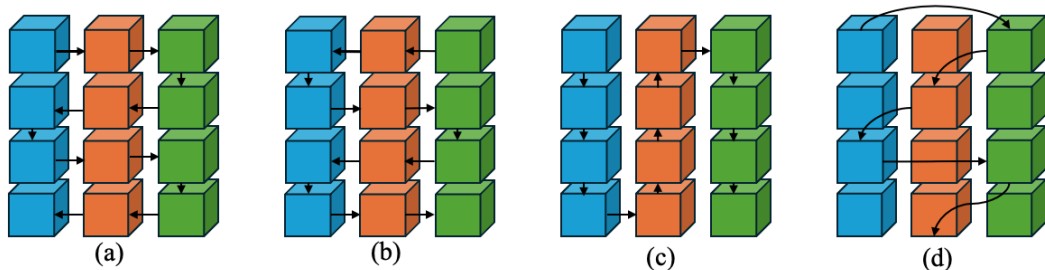

Figure 2: Illustration of four scan methods. (a) Cross-dimension scan captures interactions across different dimensions. (b) Bidirection-dimension scan processes each dimension in both directions, ensuring comprehensive feature extraction along each axis. (c) Inner-dimension scan focuses on sequential patterns within individual dimensions, such as temporal trends or stock-specific behavior. (d) Skip-dimension scan enhances efficiency by skipping certain state space, reducing redundancy while maintaining crucial information.

$$\text{TS} = 2\text{DScan}_{\text{Time}}(1\text{DScan}_{\text{Time}}(\tilde{x}_{u,t}^{\text{TSSS}}))$$
$$\text{SS} = 2\text{DScan}_{\text{Stock}}(1\text{DScan}_{\text{Stock}}(\text{TS})) \qquad (6)$$
$$\text{MS} = 2\text{DScan}_{\text{Market}}(1\text{DScan}_{\text{Market}}(\text{SS}))$$

The 1D and 2D scans are applied sequentially along the time, stock, and market dimensions, progressively refining the feature representations by capturing both intra-dimensional and inter-dimensional dependencies. The output of this process, MS, encapsulates the dynamic interactions across all three dimensions. This 3D scan layer is crucial for capturing complex dependencies in stock prediction tasks, as it facilitates multi-dimensional feature extraction and enhances the model's understanding of how temporal trends, stock-specific patterns, and market conditions interact to affect stock prices. We also integrate four scanning techniques: cross-dimension, bidirection-dimension, inner-dimension, and skip-dimension scans, as illustrated in Figure 3. This approach enhances the model's ability to capture complex dependencies across multi-dimensions, while maintaining computational efficiency.

**Fusion Layer.** The fusion layer further refines the information gathered from the Time-Spectral State Space (TSSS) and the 3D Scan layer. By using the stock-specific features from the TSSS ($\tilde{x}_{u,t}^{\text{TSSS}}$) and the refined features from the 3D Scan (MS), the fusion module integrates these different representations. This allows the model to attend to critical features across both local and global contexts, enhancing the accuracy of the final prediction. Here we use a simple mlp-based fusion module for simplicity. This fusion helps the model balance short-term fluctuations with long-term market trends, ensuring that the final embedding contains the most relevant information for prediction. The fusion function can be formalized as:

$$\tilde{x}_{u,t}^{\text{Fusion}} = \text{Fusion}(\tilde{x}_{u,t}^{\text{TSSS}}, \text{MS}) \qquad (7)$$

**Prediction and Training.** After obtaining the stock embedding $\tilde{x}_{u,t}^{\text{Fusion}}$, we pass it through a linear predictor for regression, estimating the target labels. The predictions are evaluated using Mean Squared Error (MSE) loss (Feng et al., 2021). The predicted stock values $\hat{r}_u$ are computed as:

$$\hat{r}_u = g(\tilde{x}_{u,t}^{\text{Fusion}}), \qquad (8)$$

The overall loss for the batch is computed by aggregating the MSE across all stocks in the set $S$:

$$\mathcal{L} = \sum_{u \in S} \text{MSE}(r_u, \hat{r}_u), \qquad (9)$$

where $r_u$ represents the ground truth stock values, and $\hat{r}_u$ denotes the predicted values. This joint optimization ensures that the model learns to capture the shared patterns and dependencies across different stocks for accurate predictions.

## 4 EXPERIMENTS

**Datasets.** Our framework is evaluated using data from the Chinese stock market, focusing on the CSI300 and CSI800 stock indices. These indices represent the top 300 and 800 stocks by market capitalization, respectively, on the Shanghai Stock Exchange and Shenzhen Stock Exchange. The dataset includes daily records from 2008 to 2022 for both indices. We use data from Q1 2008 to Q1 2020 for training, Q2 2020 for validation, and the final ten quarters, from Q3 2020 to Q4 2022, for testing. Stock features are extracted using publicly available Alpha158 indicators (Yang et al., 2020). The lookback window $\tau$ is set to 8 days, and the prediction horizon $d$ is 5 days. Furthermore, 63 market features are constructed using CSI300, CSI500, and CSI800 indices with referable intervals $d'$ of 5, 10, 20, 30, and 60 days. In addition, we integrate subjective market insights by scraping analyst reports published one day, three days, and one week prior to each prediction date. These reports provide sentiment and market analysis, which are included as text features to enhance the model's predictive capabilities. The web scraping technique enables the collection of relevant reports from financial platforms, enriching our dataset with real-time sentiment information from industry experts.

**Baselines.** We assess the performance of OracleMamba in comparison to several stock price forecasting baselines, categorized as follows. •XGBoost (Chen & Guestrin, 2016): A decision tree-based model, recognized as one of the most robust baselines, as highlighted on the Qlib platform leaderboard (Yang et al., 2020). •LSTM (Graves & Graves, 2012), GRU (Cho, 2014), TCN (Bai et al., 2018), and Transformer (Vaswani, 2017): Sequential models that use the standard LSTM, GRU, temporal convolutional network, and Transformer architectures to forecast stock prices over time. •GAT (Veličković et al., 2017): A graph-based approach where stock representations are initially captured using a sequential encoder, followed by information aggregation through graph attention networks. •DTML (Yoo et al., 2021): A state-of-the-art technique for mining stock correlations, leveraging an attention mechanism to discover dynamic relationships among stocks while incorporating market data into the model. •MASTER (Li et al., 2024a): A cutting-edge stock price forecasting model designed to address market volatility and complex correlations. It utilizes a market-index-guided gating mechanism for precise feature selection, combined with a Transformer-based architecture to improve predictive accuracy and adaptability.

**Evaluation.** We employ both ranking and portfolio-based metrics to comprehensively evaluate model performance. For ranking metrics, we consider four key measures: Information Coefficient (IC), Rank Information Coefficient (RankIC), Information Ratio-adjusted IC (ICIR), and Information Ratio-adjusted RankIC (RankICIR). IC and RankIC represent the daily-averaged Pearson and Spearman correlations, respectively, while ICIR and RankICIR are their normalized counterparts, calculated by dividing the IC and RankIC by their standard deviations. These metrics are commonly used in the literature (Xu et al., 2021; Yang et al., 2020) to assess forecasting performance from both a value-based and ranking perspective. Additionally, we evaluate investment performance using two portfolio-based metrics that capture both profitability and risk. Using a straightforward strategy of selecting the top 30 stocks based on their return ratios, we report the Excess Annualized Return (AR) and Information Ratio (IR). AR reflects the expected annualized excess return, while IR measures the risk-adjusted performance of the portfolio. Then we conduct two-tailed t-tests (Liu et al., 2019) to compare the performance.

**Implementation.** We implemented OracleMamba using PyTorch, and developed our methods on top of the open-source quantitative investment platform Qlib (Yang et al., 2020). For DTML, we followed the original paper for implementation, as no official version is publicly available. For other baselines, we utilized their respective Qlib implementations. The hyperparameters of each baseline method, including the number of layers and model size, were tuned from the sets $\{1, 2, 3\}$ and $\{128, 256, 512\}$ respectively. The learning rate, lr, was tuned among $\{10^{-i}\}_{i \in \{3,4,5,6\}}$, and the best hyperparameters were selected based on the IC performance during the validation phase. For MASTER (Li et al., 2024a), we tuned the model size $D$ and learning rate lr within the same ranges as the baselines. The final selection was $D = 256$, lr $= 10^{-5}$ for all datasets, and we set $N_1 = 4$, $N_2 = 2$ for all datasets, with $\beta = 5$ for CSI300 and $\beta = 2$ for CSI800. Each model was trained for up to 40 epochs with early stopping. All experiments were conducted on a server equipped with a single GPU (NVIDIA GeForce RTX 3090). Each experiment was repeated five times with random initialization, and the average performance was reported.

Table 1: Overall performance comparison in the four datasets. The t-test results show that our performance advantage over the previous SOTA method is statistically significant Liu et al. (2019) ($p < 10^{-2}$). $\star$: $p < 10^{-2}$, $\star\star$: $p < 10^{-4}$.

| Dataset | Model | IC | ICIR | RankIC | RankICIR | AR | IR |
|---------|-------|-----|------|--------|----------|-----|-----|
| CSI300 | XGBoost | $0.051 \pm 0.001$ | $0.37 \pm 0.01$ | $0.050 \pm 0.001$ | $0.36 \pm 0.01$ | $0.23 \pm 0.03$ | $1.9 \pm 0.3$ |
| | LSTM | $0.049 \pm 0.001$ | $0.41 \pm 0.01$ | $0.051 \pm 0.002$ | $0.41 \pm 0.03$ | $0.20 \pm 0.04$ | $2.0 \pm 0.4$ |
| | GRU | $0.052 \pm 0.004$ | $0.35 \pm 0.04$ | $0.052 \pm 0.005$ | $0.34 \pm 0.04$ | $0.19 \pm 0.04$ | $1.5 \pm 0.3$ |
| | TCN | $0.050 \pm 0.002$ | $0.33 \pm 0.04$ | $0.049 \pm 0.002$ | $0.31 \pm 0.04$ | $0.18 \pm 0.05$ | $1.4 \pm 0.5$ |
| | Transformer | $0.047 \pm 0.007$ | $0.39 \pm 0.04$ | $0.051 \pm 0.002$ | $0.42 \pm 0.04$ | $0.22 \pm 0.06$ | $2.0 \pm 0.4$ |
| | GAT | $0.054 \pm 0.002$ | $0.36 \pm 0.02$ | $0.041 \pm 0.002$ | $0.25 \pm 0.02$ | $0.19 \pm 0.03$ | $1.3 \pm 0.3$ |
| | DTML | $0.051 \pm 0.001$ | $0.37 \pm 0.01$ | $0.050 \pm 0.001$ | $0.36 \pm 0.01$ | $0.23 \pm 0.03$ | $1.9 \pm 0.3$ |
| | LSTM | $0.049 \pm 0.006$ | $0.33 \pm 0.04$ | $0.052 \pm 0.005$ | $0.33 \pm 0.04$ | $0.21 \pm 0.03$ | $1.7 \pm 0.3$ |
| | MASTER | $0.064 \pm 0.006$ | $0.42 \pm 0.04$ | $0.076 \pm 0.005$ | $0.49 \pm 0.04$ | $0.27 \pm 0.05$ | $2.4 \pm 0.4$ |
| | **OracleMamba** | $\mathbf{0.079^{\star\star} \pm 0.003}$ | $\mathbf{0.60^{\star\star} \pm 0.02}$ | $\mathbf{0.093^{\star\star} \pm 0.003}$ | $\mathbf{0.65^{\star\star} \pm 0.04}$ | $\mathbf{0.43^{\star\star} \pm 0.04}$ | $\mathbf{3.7^{\star\star} \pm 0.2}$ |
| CSI800 | XGBoost | $0.040 \pm 0.000$ | $0.37 \pm 0.01$ | $0.047 \pm 0.000$ | $0.42 \pm 0.01$ | $0.08 \pm 0.02$ | $0.6 \pm 0.2$ |
| | LSTM | $0.028 \pm 0.002$ | $0.32 \pm 0.02$ | $0.039 \pm 0.002$ | $0.41 \pm 0.03$ | $0.09 \pm 0.02$ | $0.9 \pm 0.2$ |
| | GRU | $0.039 \pm 0.002$ | $0.36 \pm 0.05$ | $0.044 \pm 0.003$ | $0.39 \pm 0.07$ | $0.07 \pm 0.04$ | $0.6 \pm 0.3$ |
| | TCN | $0.038 \pm 0.002$ | $0.33 \pm 0.04$ | $0.045 \pm 0.002$ | $0.38 \pm 0.05$ | $0.05 \pm 0.04$ | $0.4 \pm 0.3$ |
| | Transformer | $0.040 \pm 0.003$ | $0.43 \pm 0.03$ | $0.048 \pm 0.003$ | $0.51 \pm 0.05$ | $0.13 \pm 0.04$ | $1.1 \pm 0.3$ |
| | GAT | $0.043 \pm 0.002$ | $0.39 \pm 0.02$ | $0.042 \pm 0.002$ | $0.35 \pm 0.02$ | $0.10 \pm 0.04$ | $0.7 \pm 0.3$ |
| | DTML | $0.039 \pm 0.004$ | $0.29 \pm 0.03$ | $0.053 \pm 0.008$ | $0.37 \pm 0.06$ | $0.16 \pm 0.03$ | $1.3 \pm 0.2$ |
| | MASTER | $0.052 \pm 0.006$ | $0.40 \pm 0.06$ | $0.066 \pm 0.007$ | $0.48 \pm 0.06$ | $0.28 \pm 0.02$ | $2.3 \pm 0.3$ |
| | **OracleMamba** | $\mathbf{0.087^{\star\star} \pm 0.003}$ | $\mathbf{0.98^{\star\star} \pm 0.04}$ | $\mathbf{0.096^{\star\star} \pm 0.003}$ | $\mathbf{0.81^{\star\star} \pm 0.02}$ | $\mathbf{0.52^{\star\star} \pm 0.02}$ | $\mathbf{4.5^{\star\star} \pm 0.1}$ |

**Overall Performance**

The overall performance is presented in Table 1. OracleMamba demonstrates the best results across all ranking metrics and consistently outperforms all benchmarks in portfolio-based metrics compared to the second-best results on average. Specifically, OracleMamba shows a 30% improvement in ranking metrics on CSI300, an 82% improvement on CSI800, a 57% improvement in portfolio-based metrics on CSI300, and a 91% improvement on CSI800 over the second-best results. It is important to note that ranking metrics are calculated across the entire stock set, while portfolio-based metrics primarily focus on the top 30 performing stocks. These strong results in both types of metrics suggest that OracleMamba excels at predicting the entire stock set without compromising accuracy for the most important stocks. The significant improvements highlight the critical role of stock correlation modeling, enabling each stock to benefit from the historical signals of other momentarily correlated stocks. Interestingly, previous SOTA models tended to perform better on CSI800 than on CSI300, likely due to the larger market capitalization of companies in CSI300, whose stock prices are more predictable (Li et al., 2024a). However, with our model, this trend is reversed. This suggests that our approach is particularly effective at capturing the dynamics of larger, more complex stocks and inter-stock correlation in CSI800, demonstrating its robustness across varying market conditions.

**OracleMamba Architecture**

We evaluate the effectiveness of our specialized stock transformer architecture through experiments across eight configurations: (1) OracleMamba$_{NG}$: OracleMamba without gate, where the gating mechanism is removed from our stock transformer; (2) Bi-Transformer: Two-layer Transformer, where the single-layer transformer encoder is replaced with a two-layer version to demonstrate that the model's effectiveness is not solely dependent on encoder depth; (3) Tri-Transformer: Three-layer Transformer, which further extends the encoder to three layers to explore deeper architectures; and (4) SSM, a sequential stock model without transformers, used to assess the contribution of the transformer structure itself. Additionally, we include comparisons with (5) Mamba (Gu & Dao, 2023), (6) Mamba-2 (Dao & Gu, 2024), (7) TransMamba: Transformer+Mamba, and (8) TransMamba-2: Transformer+Mamba-2 to evaluate hybrid approaches and variations of our architecture.

All experiments were conducted on the CSI300 and CSI800 datasets. The results, shown in Table 2 and Table 3, highlight the effectiveness of our tailored OracleMamba architecture, which alternates between intra-stock and inter-stock aggregation, and effectively captures both short-term and long-term correlations across the time dimension.

Surprisingly, although the attention mechanism is widely used in deep learning models, it performs poorly when combined with the Mamba architecture. This suggests that the attention mechanism

may not be inherently compatible with Mamba, likely due to a fundamental mismatch between the two systems. Specifically, (1) the Mamba model relies on ordered sequences with strict sequential dependencies, where the state at time step $t + 1$ is determined by both the previous time step $t$ and an associated hidden state. (2) On the other hand, cross-attention mechanisms treat all tokens in a sequence equally. This difference could compromise the Mamba model's capacity to model sequences effectively, as cross-attention fails to maintain the sequential integrity and hierarchical dependencies that are critical to the model's function.

Table 2: Experiments on CSI300 to validate the effectiveness of proposed stock transformer architecture. The best results are in bold and the second-best results are underlined.

| Dataset | Model | IC | ICIR | RankIC | RankICIR | AR | IR |
|---|---|---|---|---|---|---|---|
| CSI300 | SSM | 0.040 ± 0.009 | 0.26 ± 0.07 | 0.043 ± 0.007 | 0.27 ± 0.09 | 0.10 ± 0.09 | 1.2 ± 0.7 |
| | Mamba | 0.052 ± 0.004 | 0.32 ± 0.05 | 0.057 ± 0.005 | 0.39 ± 0.05 | 0.17 ± 0.06 | 1.6 ± 0.3 |
| | Mamba-2 | 0.050 ± 0.005 | 0.33 ± 0.04 | 0.053 ± 0.007 | 0.37 ± 0.06 | 0.20 ± 0.03 | 2.2 ± 0.4 |
| | TransMamba | 0.049 ± 0.003 | 0.31 ± 0.04 | 0.038 ± 0.006 | 0.23 ± 0.04 | 0.15 ± 0.06 | 1.2 ± 0.4 |
| | TransMamba-2 | 0.057 ± 0.007 | 0.28 ± 0.05 | 0.048 ± 0.006 | 0.31 ± 0.05 | 0.16 ± 0.04 | 1.5 ± 0.3 |
| | Bi-Transformer | 0.053 ± 0.004 | 0.27 ± 0.09 | 0.054 ± 0.009 | 0.35 ± 0.07 | 0.14 ± 0.06 | 1.4 ± 0.8 |
| | Tri-Transformer | 0.057 ± 0.006 | 0.31 ± 0.06 | 0.068 ± 0.009 | 0.41 ± 0.08 | 0.19 ± 0.09 | 2.0 ± 0.6 |
| | OracleMamba$_{NG}$ | **0.077** ± **0.003** | **0.56** ± **0.03** | **0.087** ± **0.003** | **0.63** ± **0.03** | **0.41** ± **0.05** | **3.3** ± **0.2** |

Table 3: Experiments on CSI800 to validate the effectiveness of proposed stock transformer architecture. The best results are in bold and the second-best results are underlined.

| Dataset | Model | IC | ICIR | RankIC | RankICIR | AR | IR |
|---|---|---|---|---|---|---|---|
| CSI800 | SSM | 0.036 ± 0.006 | 0.22 ± 0.08 | 0.035 ± 0.008 | 0.19 ± 0.08 | 0.07 ± 0.05 | 1.0 ± 0.7 |
| | Mamba | 0.050 ± 0.002 | 0.33 ± 0.04 | 0.049 ± 0.002 | 0.31 ± 0.04 | 0.18 ± 0.05 | 1.4 ± 0.5 |
| | Mamba-2 | 0.047 ± 0.007 | 0.30 ± 0.04 | 0.051 ± 0.002 | 0.42 ± 0.04 | 0.22 ± 0.06 | 2.0 ± 0.4 |
| | TransMamba | 0.043 ± 0.005 | 0.32 ± 0.06 | 0.048 ± 0.005 | 0.30 ± 0.04 | 0.16 ± 0.05 | 1.2 ± 0.5 |
| | TransMamba-2 | 0.044 ± 0.004 | 0.29 ± 0.04 | 0.051 ± 0.006 | 0.30 ± 0.05 | 0.17 ± 0.04 | 1.4 ± 0.4 |
| | Bi-Transformer | 0.045 ± 0.006 | 0.31 ± 0.05 | 0.053 ± 0.008 | 0.33 ± 0.09 | 0.18 ± 0.07 | 1.6 ± 0.3 |
| | Tri-Transformer | 0.049 ± 0.008 | 0.33 ± 0.10 | 0.059 ± 0.007 | 0.40 ± 0.07 | 0.23 ± 0.05 | 1.7 ± 0.4 |
| | OracleMamba$_{NG}$ | **0.084** ± **0.002** | **0.93** ± **0.03** | **0.092** ± **0.005** | **0.76** ± **0.04** | **0.48** ± **0.02** | **4.4** ± **0.2** |

**1D Scan, 2D Scan and 3D Scan**

In this experiment, we compare the performance of three different scanning dimensions—1DScan, 2DScan, and 3DScan—on two stock markets, CSI300 and CSI800, across six key metrics: IC, ICIR, RankIC, RankICIR, AR, and IR.

The results show a noticeable improvement in IC as the scanning dimension increases, particularly in CSI800, where 3DScan significantly outperforms both 1DScan and 2DScan, suggesting that higher-dimensional scans capture more market information, thereby enhancing predictive accuracy. Similarly, ICIR increases with scanning dimensions, with both 2DScan and 3DScan showing substantial gains over 1DScan in both markets, and CSI800 particularly benefiting from higher-dimensional scans, indicating that the enhanced information capture comes with improved stability. RankIC shows significant improvement as scanning dimensions increase, especially from 1DScan to 2DScan, while 3DScan brings CSI800 and CSI300 closer in terms of RankIC, with CSI800 showing superior performance, demonstrating a consistent improvement in ranking prediction accuracy with higher-dimensional scans. RankICIR also rises with scanning dimensions, with CSI800 showing more pronounced gains, particularly from 1D to 3DScan, suggesting that higher-dimensional scans not only enhance ranking accuracy but also reduce volatility, improving robustness. Annualized Return (AR) increases steadily with scanning dimensions, particularly in CSI800, where 3DScan shows a notable advantage, indicating that higher-dimensional scans boost returns, though the improvement in CSI300 is more modest, implying greater market stability in larger markets. Information Ratio (IR) improves with dimension, with CSI800 showing a marked increase from 2DScan to 3DScan, whereas CSI300's IR remains relatively stable across all dimensions, suggesting that higher-dimensional scans better balance risk and reward, especially in larger, more complex markets like CSI800.

Overall, higher-dimensional scans, such as 3DScan, capture more market information and significantly improve IC, RankIC, and other metrics, especially in CSI800, demonstrating that 3DScan enhances predictive accuracy, return, and stability, making it more adaptable and robust across different markets. Compared to lower-dimensional scans, higher-dimensional scans offer a better risk-

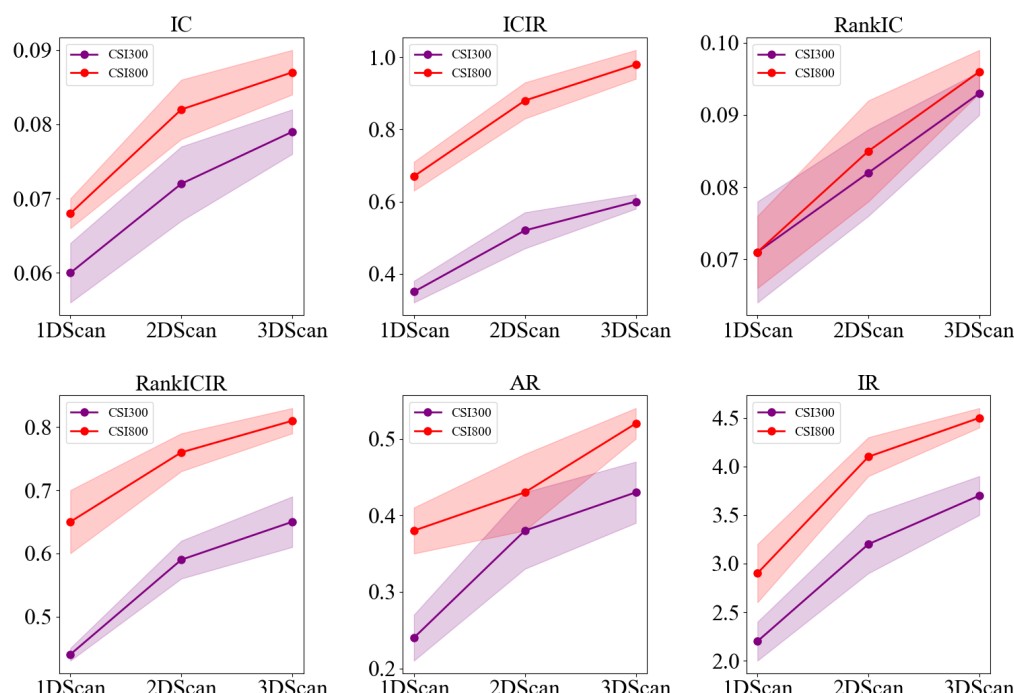

Figure 3: Performance comparison of 1D Scan, 2D Scan, and 3D Scan across the CSI300 and CSI800 datasets.

reward balance (IR), particularly in larger markets like CSI800, indicating that higher-dimensional scanning techniques are more effective at optimizing returns while controlling risk. Thus, 3DScan outperforms both 1DScan and 2DScan, particularly in complex markets like CSI800, suggesting that incorporating more dimensions leads to better market performance. Future research and applications should consider adopting higher-dimensional scanning for improved results. The results are shown in Figure 3.

## 5 CONCLUSION

We present OracleMamba, a novel stock price forecasting method that is designed to effectively capture both short-term and long-term informative correlations. It integrates two key components: a dynamic market-guided module and a SelectiveMamba module. The dynamic market-guided module enhances the model's short-term predictive capabilities by incorporating market sentiment and objective market data, allowing it to quickly respond to rapid fluctuations in stock prices. Meanwhile, the SelectiveMamba module focuses on long-term forecasting by filtering out noise and preserving key signals, which helps in building strong correlations over time. Extensive experiments on the CSI300 and CSI800 datasets demonstrate its superiority, with an average improvement of 56% in ranking metrics and 74% in portfolio-based metrics over baseline models.

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
