# OpenReview forum: "OracleMamba: A Dynamic Market-Guided and Time State Selection Framework for Robust Stock Prediction"
_ICLR.cc/2025/Conference — ICLR 2025 Conference Withdrawn Submission_

### Official Review · Reviewer_GKGo · 2024-11-02

**Soundness:** 2
**Presentation:** 2
**Contribution:** 1
**Rating:** 1
**Confidence:** 4

**Summary:**

This paper is the first to introduce Mamba for stock price forecasting, focusing on efficiently combining various short- and long-term factors necessary for stock prediction. The study aims to efficiently integrate time-series data and text data, such as analyst reports, through the dynamic market-guided module, while combining short- and long-term contexts through the SelectiveMamba module. The model was tested on Chinese stock market data, with results showing that it outperforms baseline models. However, the paper lacks a thorough literature review on stock price forecasting. Only three baseline models are referenced for comparison, two of which are studies published at least three years ago. Additionally, while the authors state they used scraped analyst reports, they fail to specify the sources of the reports, the authors of these reports, or the number of reports used, which raises concerns about the credibility of their experiment and data.

**Strengths:**

According to the authors, this is the first study to apply Mamba to stock price forecasting (although this claim is questionable due to the lack of a detailed literature review).

**Weaknesses:**

The authors claim that incorporating sentiment analysis, such as analyst reports, into stock price forecasting is novel, but many studies have already pursued this approach. Moreover, multiple studies within stock price forecasting have also explored combining short- and long-term contexts. Therefore, the two main contributions claimed by the authors are not particularly new, and the lack of a comprehensive search and mention of previous studies is a significant oversight.

Furthermore, the fact that testing was limited to the Chinese market decreases the reliability of the experimental results, and the lack of detailed information on the analyst reports, which are crucial data, also presents serious issues with the study’s credibility and reproducibility.

**Questions:**

- Many studies have already combined sentiment analysis with stock price forecasting, so why is there no mention of these studies in this paper?
- The approach of combining short- and long-term contexts has also been widely studied within the context of stock price forecasting and has been presented at major ML/AI conferences. Why wasn’t this mentioned, and what advantages does this study have compared to those prior works?

---

### Official Review · Reviewer_Ap4Z · 2024-11-03

**Soundness:** 2
**Presentation:** 3
**Contribution:** 2
**Rating:** 5
**Confidence:** 4

**Summary:**

The article includes a dynamic market-guided module and the SelectiveMamba module, effectively addressing the challenges posed by noisy data. It introduces Mamba into stock price forecasting and employs a 3D scan method. By integrating market sentiment, the article incorporates subjective factors. Experiments demonstrate its performance.

**Strengths:**

1.The article introduces Mamba into stock price forecasting.

2.The article proposes a 3D scan method to capture interactions across the dimensions of time, stock, and market state.

3.The method proposed in the article performs well in experiments.

**Weaknesses:**

1. The article lacks novelty and sufficient discussion on related work:

   a) While using GPT to process market sentiment is indeed innovative, there have been many previous works that utilized text data, including the use of pre-trained language models as in [1].

   b) It is difficult to see the relationship between the Time-Spectral method in the article and stock data. It appears to merely apply a time series method to stock data without analyzing how the Time-Spectral method aids stock prediction or offers any special improvements for it. Additionally, frequency methods are also common in time series analysis, as in [2].

   c) Inter-stock correlations have been used in many previous articles, as in [3].

2. The article claims that the model can handle noisy data, but I do not see how or why noisy data can be addressed, or what improvements have been made compared to previous methods. Why could previous deep learning methods not handle noisy data while the current one can? Furthermore, noise in stock data is often caused by random Brownian motion, which the article does not analyze or explain theoretically.

3. The article mentions the application of the Mamba model, but the method section does not clearly indicate where it is used for those unfamiliar with the Mamba model. Nor does it explain how the Mamba model contributes to stock prediction.

4. The baselines are relatively weak, with only two methods specifically tailored for stock data.

5. The article does not provide the code.


[1] Yang, Linyi, et al. "Numhtml: Numeric-oriented hierarchical transformer model for multi-task financial forecasting." Proceedings of the AAAI Conference on Artificial Intelligence. Vol. 36. No. 10. 2022.

[2] Liu, Shengzhong, et al. "FOCAL: Contrastive learning for multimodal time-series sensing signals in factorized orthogonal latent space." Advances in Neural Information Processing Systems 36 (2024).

[3] Cheng, Rui, and Qing Li. "Modeling the momentum spillover effect for stock prediction via attribute-driven graph attention networks." Proceedings of the AAAI Conference on artificial intelligence. Vol. 35. No. 1. 2021.

**Questions:**

Please refer to the "Weaknesses" for your response, especially the first three points.

---

### Official Review · Reviewer_T7mb · 2024-11-03

**Soundness:** 2
**Presentation:** 2
**Contribution:** 2
**Rating:** 5
**Confidence:** 4

**Summary:**

This paper presents a Mamba-based framework for stock return prediction by leveraging financial market data, such as stock prices, market indices, and market sentiment.

**Strengths:**

1. The authors proposed a novel 3D scanning mechanism for analyzing financial market information
2. The problem to be solved is well formulated

**Weaknesses:**

1. The data used in the paper is not clearly specified
2. The motivation behind the SSM design is not well explained
3. The experiment lacks comparisons with some important baselines

**Questions:**

The market state encoding section lacks sufficient details to justify the use of market state information:
1. The authors failed to describe the subjective context thoroughly. There is only a vague mention of analysts' reports and financial documents scraped from unspecified platforms. It is unclear what these documents are, their market coverage, their frequency, and the volume of data involved. This information is crucial. For example, if only quarterly updated earnings reports from some public companies are used, how do they align with daily updated stock prices? Additionally, how noisy is this data? The current version of the paper is problematic and does not justify the use of market context.
2. It is unclear how the GPT-O1 model is used to convert these textual data into sentiment. The prompts used are not described. The authors did not specify what the sentiment results look like. Are they presented as sentiment scores, sentiment labels, etc.?
3. The authors should more clearly specify the role of the experts. Do they mean that the experts are the ones who wrote the documents, or are they directly involved in the analysis?
4. It is unclear how sectors and regions are processed and embedded in the data or how they are used. The authors seem to only mention these aspects without integrating them into their analysis.

The TSSS structure has several issues:
1. Why are B and C input-independent? This is more like a basic SSM structure that is time-invariant, while Mamba is designed to be an improvement on such structures.
2. What is $s$ in the calculation of DSE?
3. The design of DTE and DSE lacks explanation, and it is unclear how they reflect the benefits claimed by the authors in the relevant section.

Experiment Section:
1. The setup for the comparison methods is unclear. Are these methods using the same data as the proposed model, or are they only using stock price data?
2. Many SOTA time series forecasting models are not included in the comparison, such as DLinear, NLinear, Autoformer, Fedformer, PatchTST, etc.
3. Why was the vanilla Transformer model used for comparison when there are many Transformer-based models specifically designed for time series tasks?
4. Comparisons between the proposed model and Mamba-based models (or Mamba itself) are needed.

---

### Official Review · Reviewer_VN4H · 2024-11-04

**Soundness:** 2
**Presentation:** 2
**Contribution:** 2
**Rating:** 5
**Confidence:** 4

**Summary:**

The paper presents OracleMamba, a new stock prediction framework designed to integrate both short-term and long-term market dynamics using a dynamic market-guided module and a SelectiveMamba module. The model aims to address the limitations of previous joint forecasting models by effectively balancing short-term market volatility and long-term trends. OracleMamba uses a comprehensive market-guided gating mechanism that fuses market sentiment and objective market indicators to enhance prediction accuracy, while the SelectiveMamba module captures spectral and temporal features to reduce noise and extract key signals from market data.

**Strengths:**

1. This paper studies the first work that introduces Mamba into stock price forecasting, which could be promising for the development of this subdomain.
2.  The integration of a market-guided module for short-term forecasting and a SelectiveMamba module for long-term stability represents an novel hybrid approach to stock prediction.
3. The paper is written with good clarity and thus is easy to follow.

**Weaknesses:**

1. A key concern is the lack of clarity regarding whether the additional information used to enhance OracleMamba is also used for baseline comparison. It is also unclear how much these features specifically contribute to OracleMamba's performance. If baseline models do not incorporate the same information, the comparison may be unfair.
2. The exact data point length in CSI300 and CSI800 for model training is not specified. Is the data daily-based or hourly-based? Given that Mamba-based methods are used, a longer financial data sequence might provide an advantage.
3. Since the paper adopts a Mamba-based solution, it is crucial to evaluate the computational cost, including memory usage and runtime efficiency.

**Questions:**

Please see the weakness for details.

---

### Note · Authors · 2025-03-19

I have read and agree with the venue's withdrawal policy on behalf of myself and my co-authors.

---

### Meta-Review · Area_Chair_78bY · 2024-12-16

**Metareview:**

The paper for the first time proposed to introduces Mamba into stock price forecasting, which is novel. The problem is well formulated and the paper is clearly presented. Experiment results show the effectiveness of the proposed model. However, the reviewers have also raised many issues that the paper needs to be further improved, including the limited novelty, the insufficient experiment and discussions, and the unclear explanation on the motivation.

**Additional Comments On Reviewer Discussion:**

There is no author rebuttal and reviewer discussion.

---

### Decision · Program_Chairs · 2025-01-22

Reject